# Artificial Intelligence (AI) versus POCUS Expert: A Validation Study of Three Automatic AI-Based, Real-Time, Hemodynamic Echocardiographic Assessment Tools

**DOI:** 10.3390/jcm12041352

**Published:** 2023-02-08

**Authors:** Eyal Gohar, Amit Herling, Mor Mazuz, Gal Tsaban, Tomer Gat, Sergio Kobal, Lior Fuchs

**Affiliations:** 1Ben-Gurion University of the Negev, Beer Sheva 8410501, Israel; 2Soroka Medical Center, Beer Sheva 84101, Israel

**Keywords:** Point Of Care UltraSound (POCUS), Ejection Fraction (EF), Velocity Time Integral (VTI), Inferior Vena Cava measurement, IVC collapsibility index (IVC CI), Intensive Care Unit (ICU), Real Time US tool, Artificial Intelligence, Echocardiography, Ben Gurion University of the Negev, Soroka University Medical Center

## Abstract

Background: Point Of Care Ultra-Sound (POCUS) is an operator dependent modality. POCUS examinations usually include ‘Eyeballing’ the inspected anatomical structure without conducting accurate measurements due to complexity and insufficient time. Automatic real time measuring tools can make accurate measurements fast and simple and dramatically increase examination reliability while saving the operator much time and effort. In this study we aim to assess three automatic tools which are integrated into the Venue™ device by GE: the automatic ejection fraction, velocity time integral, and inferior vena cava tools in comparison to the gold standard—an examination by a POCUS expert. Methods: A separate study was conducted for each of the three automatic tools. In each study, cardiac views were acquired by a POCUS expert. Relevant measurements were taken by both an auto tool and a POCUS expert who was blinded to the auto tool’s measurement. The agreement between the POCUS expert and the auto tool was measured for both the measurements and the image quality using a Cohen’s Kappa test. Results: All three tools have shown good agreement with the POCUS expert for high quality views: auto LVEF (0.498; *p* < 0.001), auto IVC (0.536; *p* = 0.009), and the auto VTI (0.655; *p* = 0.024). Auto VTI has also shown a good agreement for medium quality clips (0.914; *p* < 0.001). Image quality agreement was significant for the auto EF and auto IVC tools. Conclusions: The Venue™ show a high agreement with a POCUS expert for high quality views. This shows that auto tools can provide reliable real time assistance in performing accurate measurements, but do not reduce the need of a good image acquisition technique.

## 1. Background

Point Of Care Ultra-Sound (POCUS) is a modality that uses the Ultrasound device as an adjunct to the physical examination. The Ultrasound is operated bedside and is intended to supply the clinician with fast, accurate, clinically relevant information the clinician could then use for the follow-up, diagnosis, or treatment of the patient [1,2]. POCUS is an operator-dependent modality. Currently, a POCUS examination quality relies to a great extent on the operator’s technique and image acquisition skill, and on the operator’s ability to accurately interpret the sonographic views obtained.

POCUS image interpretation is often performed by ‘eyeballing’ the acquired image without using accurate measurements [3,4]. This creates a dependence upon the operators’ experience and image interpretation abilities and subjects the examination to inaccuracies and inter-examiner variance.

In recent years, POCUS has become a significant part of many clinicians’ daily practice [5,6]. Several guidelines recommend using POCUS for procedural guidance, hemodynamic and diagnostic assessment [7,8]. The use of POCUS has expanded further during the Coronavirus-disease-19 pandemic due to the need for accurate, objective, and time-efficient patient assessment, increased physician safety, and easier logistics [9]. POCUS is particularly efficient in the intensive care setting and has a significant impact in optimizing critical care [6] and in the treatment and differentiation of shock [10].

One of the ways to reduce testing variation is the implementation of reliable artificial intelligence real-time automatic tools for improving the interpretation of POCUS examinations without the need for human assessment or manual measurements. Reducing the dependence on operators’ abilities can make the information obtained more reliable and suitable for decision making. Moreover, manual accurate measurements are often complicated and time-consuming [11]. For example, the Simpson method for measuring EF requires the operator to manually trace the endocardial border in systole and diastole in at least two views. Such accurate and time-efficient real-time automatic tools could improve urgent decision making and, therefore, patient care. Lastly, automatic quality feedback (this feature is integrated into all three tools) and instantaneous measurable feedback can augment novice POCUS self-confidence and their teaching process.

Left ventricular (LV) Ejection Fraction is an important clinical index. EF assessment can be extremely valuable in evaluating hemodynamically unstable patients [12] and dyspneic patients [4]. It can also diagnose heart failure in different settings, such as the emergency department and the internal wards [13,14]. The classic method for measuring EF is the Simpson method, which requires manually tracing the endocardial border of the left Ventricle in systole and diastole in both the apical four-chamber view (Figure 1a) and the apical two-chamber view (Figure 1c) [15].

Inferior vena cava measurements are also of high clinical yield. The central venous pressure (CVP) can be estimated by measuring the diameter of the inferior vena cava (IVC) and the collapsibility of the IVC during inspiration, known as collapsibility or Caval Index (CI) [16]. co-factoring the CI with the IVC diameter provides an assessment of the central venous pressure (CVP) [17,18]. Determining the CVP can assist in differentiate a cardiac and an obstructive shock from a septic or a hypovolemic shock [19]. In mechanically ventilated patients, the IVC distensibility is a dynamic index that can assist in identifying volume responsiveness and can help determine adequate fluid management [20,21]

The Velocity Time Integral (VTI) is classically measured at the apical five-chamber view (Figure 1b) by measuring the flow velocity through the left ventricle outlet (LVOT), which provides a velocity over time curve. Multiplying the VTI times the left ventricular outlet area equals the cardiac stroke volume, and if multiplied by the heart rate—the cardiac output [22]. The cardiac stroke volume aids in the differentiation of shock, shock treatment, determining fluid responsiveness, and in risk stratification [23,24]

All three tools can be integrated in the assessment of patients in shock, hemodynamic instability, dyspnea, and during acute care. In this study, we aim to assess the three automatic tools available on the GE Venue™ device. An automatic Real-Time Left Ventricular Ejection Fraction tool, an automatic Inferior Vena Cava measuring tool, and an automatic Velocity Time Integral measuring are integrated into the Venue™ US device and, with a push of a button, automatically recognize the relevant anatomical structures and conduct the relevant measurements. Besides measurements, all automated tools examined in this study rate the quality of the views obtained by the operator on a scale from 0–2 (presented as a red-yellow-green scale to the operator).

This is a validation study for these real-time, automatic hemodynamic assessment echocardiographic tools. Our primary objective in this study is to validate the accuracy of each automatic tool, blindly comparing the AI tool measurements to hand-measured results conducted by a POCUS expert (the gold standard). A secondary objective is validating each tool’s automatic AI quality indicator by comparing the AI tools’ quality assessments to quality assessments given by the Expert.

## 2. Methods

Three separate studies were conducted to assess each tool. All studies share a common methodology; views were acquired by an expert POCUS operator and scored by both the automatic tool (a three-level quality indicator is part of all three tools) and the operator as one of three imaging qualities. Poor quality clips (with an expert score of 0 out of 0–2) were excluded from the study. Once coded, measurements were made by both the automatic tool and the POCUS expert, who was blinded to the tool’s measurements. The agreement between the operator and the automatic tools’ results was then statistically measured to determine the reliability of the automatic tool. All examinations were performed using a Venue go™ ultrasound machine by General Electric. The POCUS expert was an Internal Medicine attending with Eight years of POCUS experience, certified by the American National Board of Echocardiography.

Patients were enrolled from the internal medicine wards at Soroka University Medical Center (SUMC) in Beer Sheva between October 2020 and May 2021. Inclusion criteria included admission to the Internal Ward, the ability to give informed consent, and no contraindication for the Echo study (chest deformation, scars on the chest, or patient discomfort during the study). Ethical approval number 0050-17-SOR was obtained by the Hospital’s local ethics committee.

## 3. Real-Time EF Measurement

A POCUS expert acquired apical four-chamber view (A4CV) clips by placing the probe horizontally over the apex of the heart (Figure 1a) and prospectively recording for at least 10 s. All clips were reviewed and blindly assessed for LVEF quantification by the Expert without exposure to the automatic tool results. Post hoc automatic analyses were performed and documented using the Venue™ real-time automatic LVEF. Each clip was considered an independent observation point.

### Left Ventricle EF Automatic Measuring Tool

Once the User scans the A4CH view, the Real-Time EF tool identifies the view using Artificial Intelligence (AI) and Machine Learning (ML) algorithms. The semi-automatic tool traces the Ventricle walls per frame and identifies end diastolic and end systolic frames based on the maximal and minimal volumes measured in each heart cycle (Figure 2). Additionally, the tool calculates the image quality based on scanning quality, the tool’s identification of the A4CH view, and the consistency of the EF results. The quality indicator is reflected in the color of the contour. Additionally, if the A4CH view is not detected for more than a few seconds, the tool can indicate the expected location of the LV on the image to the User. Once the User freezes the image, the tool enables quick navigation between the acquired heart cycles and end diastolic and end systolic frames in the last 4 s. This allows for rapid review and selection of the preferable cycle to store and document.

The auto LVEF tool provides two outputs—a score between 0–2 represented to the operator as a colored image quality marker (Red is poor = 0, Yellow is moderate = 1, Green is good = 2), and the EF assessment (presented as a percentage). The expert graded the views blindly, using the same score of 0–2 as the automatic tool for image quality (0—cannot comment on EF, 1—can estimate EF but endocardial border not fully demonstrated, 2—endocardial border well demonstrated), then estimated the EF by the eyeballing technique. We divided the LVEF expert’s assessment into three categories of LV function (preserved, mildly reduced, and reduced EF) as defined in the 2021 ESC Heart Failure guidelines for LVEF: (1) ≥50%, (2) 41–50%, and (3) ≤40%, respectively.

## 4. The IVC Collapsibility Measurement

The IVC clips were taken from the subcostal IVC view (Figure 3). In this view, the maximum and minimum diameters of the IVC throughout a respiratory cycle were measured two cm from the IVC-right atrium junction, using M-mode, providing the CI that represents the collapsibility of the IVC.

Following imaging acquisition, a post hoc analysis of the acquired IVC clip was performed in two different manners: (1) Using a real-time, machine-integrated, automatic IVC tool, and (2) by the expert physician, blinded to the automatic assessment, who examined the same clip frame-by-frame and measured the CI manually. The CI is presented as a percentage of the IVC collapsibility (minimal IVC size divided by the maximal IVC size).

Each IVC collapsibility measurement included the calculation of the Caval Index (CI), which is calculated as follows:$$\mathrm{CI}=\frac{\mathrm{maximal}\;\left(\mathrm{expiratory}\right)\;\mathrm{diameter}+\mathrm{minimal}\;\left(\mathrm{inspiratory}\right)\;\mathrm{diameter}}{\mathrm{maximal}\;\left(\mathrm{expiratory}\right)\;\mathrm{diameter}}$$

As mentioned, all clips were scored manually based on the quality of imaging from zero to two: 0—low, or failed attempt to track the IVC, or image that could not be analyzed, 1—CI can be measured and calculated but with moderate imaging quality and 2—for good quality clips with a clearly demonstrated IVC.

### IVC CI Automatic Measuring Tool

The auto IVC tool automatically places the M-mode cursor 2–3 cm below the diaphragm. The algorithm then measures the maximum and minimum diameters of the IVC throughout a respiratory cycle (Figure 4). The auto IVC tool measures the maximum and minimum diameters of the IVC throughout the respiratory cycles in real time. The collapsibility index represents the collapsibility (diameter change) of the inferior vena cava between expiration and inspiration. The quality indicator is represented by the color of the M-mode cursor, and varies between green/yellow/red to represent excellent/average/unacceptable image quality, respectively.

## 5. VTI Measurement

An expert POCUS operator acquired the views by placing the probe horizontally in the apex of the heart and prospectively recording for at least 10 s in the apical five chambers view (Figure 1b). Half of the clips were scored by the physician to quality distribution from zero to two: 0—low for a failed attempt to obtain a proper reading of the VTI or image that could not be analyzed, 1—medium for VTI can be calculated but with moderate imaging quality and 2—high for good quality clips with a clearly demonstrated VTI. The clips were then blindly measured, tracing the LVOT velocity curve manually for VTI. The automatic tool measured the same clip, which similarly scored the clips according to imaging quality. High- or medium-quality clips were then further assessed and automatically calculated for LVOT VTI values. Each clip was considered an independent observation point.

### VTI Automatic Measuring Tool

The auto VTI tool is based on proprietary artificial intelligence. The tool automatically places the cursor on the left ventricular outflow tract (LVOT) and auto-traces the velocity waveforms to calculate the VTI. The tool then calculates VTI and CO by averaging all Doppler waveforms during a period of four seconds. Calculations were performed in real-time, and the results are displayed in the Results Box (Figure 5). The Quality Indicator is represented by the color of the curser placed by the system over the LVOT. It varies between green/yellow/red to represent excellent/average/unacceptable image quality, respectively.

## 6. Statistical Analysis

To assess the agreement between the automatic and physician-assessed quantifications, we performed a Cohen’s Kappa test. A two-sided *p*-value ≤ 0.05 was considered statistically significant. To assess the agreement between the automatic and physician-assessed View Quality, we performed a Fisher’s exact test. A two-sided *p*-value ≤ 0.05 was considered statistically significant. All analyses were performed using SPSS 26.0 (Armonk, NY, USA).

## 7. Results

### 7.1. LVEF

One hundred and thirty-two LVEF clips from 44 patients were acquired from the A4C view. The clips were taken in pairs—a POCUS expert acquired a clip, and then without changing the probe position acquired a second, similar clip. Thus, two groups of paired clips were formed. All clips were interpretable and included in the analysis. Patients were primarily males, with a mean age of 55 ± 19.8 and a mean body mass index (BMI) of 26.8 ± 5.3 (Table 1). Of the first group of clips assessed by the Expert (n = 66), 53 observations were classified as preserved EF (group 1), 11 as moderately reduced EF (group 2), and 2 as severely reduced EF (group 3). High agreement (weighted Cohen’s-Kappa 0.460; *p* < 0.001) was found between the AI-based automatic EF tool and the Expert’s assessment when the test was conducted on high-quality clips (quality score of 2 as graded by the expert). A much lower agreement was observed in poor-quality clips (Table 2). Higher agreement between the automatic tool and physician EF assessment (weighted Cohen’s-Kappa 0.54; *p* < 0.001) was observed when LVEF was divided into two groups (1) ≥50% and (2) <50%. The overall agreement between the automatic tool and the physician’s EF assessment was high (weighted Cohen’s-Kappa 0.498; *p* < 0.001, Table 2). For the image quality agreement, we found a significant difference (*p* = 0.001, Fishers Exact test value—11.71, α) in the agreement of quality between the physicians’ group and the automatic tools group.

### 7.2. IVC

Forty-six IVC clips acquired from 37 patients were included in the analysis. Patients were primarily males (62%), with a mean age of 60.45 ± 15.61 and a mean body mass index (BMI) of 26.98 ± 5.48 (Table 1). The mean collapsibility index (CI) was 0.30 when measured manually by an expert and 0.35 when measured by the automatic tool. The Intraclass Correlation Coefficient for agreement between the automatic and physician-assessed quantifications was low (kappa 0.388; *p* = 0.041). When analyzed from only high-quality acquired clips (grade 2), the Intraclass Correlation Coefficient rose to 0.536 (*p* = 0.009). When analyzed from only lower and medium quality acquired clips, the Intraclass Correlation Coefficient was negative (−1.314). When divided into two groups of collapsibility index, below and above 0.5, the Intraclass Correlation Coefficient for agreement between the automatic and physician-assessed quantifications was 0.536 (95% CI 0.009 sig) (Table 3). For the image quality agreement, we found a significant difference (*p* = 0.001, Fishers Exact test value—14.311, α) in the agreement of quality between the physicians’ group and the automatic tools group.

### 7.3. LVOT VTI

One hundred and fourteen apical five-chamber view clips were acquired from 32 patients. The clips were taken in pairs—a POCUS expert acquired a clip, and then without changing the probe position acquired a second, similar clip. Thus, two groups of paired clips were formed. Patients were primarily males (69%), with a mean age of 57.4 ± 18.5 and a mean body mass index (BMI) of 27.1 ± 5.5 (Table 1). When assessed by the Expert, the mean LVOT VTI value was 19.5 ± 4.5 and 17.7 ± 4.3 by the automatic analysis read. The Intraclass Correlation Coefficient for agreement between the automatic and physician-assessed quantifications was 0.825 (95% CI 0.659, 0.905; *p* < 0.001). In the high clips’ quality subgroup, the Intraclass Correlation was reduced to 0.655 (95% CI 0.013, 0.877; *p* = 0.024). In the medium clip quality group, the Intraclass Correlation was 0.914 (95% CI −0.077, 0988; *p* < 0.001) (Table 4). For the image quality agreement, we found a significant difference (*p* = 0.167, Fishers Exact test value—3.26, α) in the agreement of quality between the physicians’ group and the automatic tools group.

## 8. Discussion

The automatic tools inspected in this study are aimed to be used bedside in a clinical setting. To determine the true feasibility of implementing the automatic tools into everyday clinical use, they need to be assessed on real patients in real clinical settings. Therefore, we believe that automatic tools designed to be used by clinicians should be validated by the users, not only by the FDA or the industry. Therefore, we attempted to validate the tools methodologically and scientifically in a real-life environment. In these three AI-based automatic tools we studied, we have found that the machine-integrated Venue™ auto real-time measurement tools are reliable when the quality of the acquired clip is good.

The Left Ventricle Ejection Fraction Automated Measuring Tool was reliable when good cardiac imaging quality is obtained. Most of the study population were patients with preserved LVEF, where the automatic tool was found to be especially reliable. We have found that the overall agreement between the Expert and machine is high (0.498; *p* < 0.001) and even higher for expert assessment of normal LV. We can confidently confirm that the tool is reliable when the LV is well demonstrated, and its function is normal. Unfortunately, our cohort did not include enough moderate and severe left ventricular systolic function to examine the tool’s accuracy when the heart is not contracting well. Although our findings are limited to normal LV systolic function and should be further studied in different levels of LV function, we believe that novice cardiac ultrasound operators can use this tool for reassurance and confirmation of their EF assessment, especially for normal LV systolic function.

For IVC size and collapsibility index assessment, the tool was found to be very reliable with an agreement coefficient of 0.536 (95% CI, *p* = 0.009) when utilized on good IVC image quality and unreliable when utilized on medium and low image quality (Table 3). This is probably due to the technical challenge of measuring the diameter of the IVC when the vein borders are not clearly demonstrated and when there is variability in the placement of the M-mode cursor (Figure 4). We confirm that when the image quality of the IVC is good, this tool is reliable, and the automatic assessment of the IVC CI can be used.

The VTI agreement proved to be very high in all levels of the acquired image quality; high, medium, and total. This finding confirms that when a quick measurement for stroke volume is needed, there is a reliable automatic tool for any level of imaging quality.

We have shown that the three described new AI-based automatic tools designed for better assessment of shock, hemodynamic instability, and evaluation of volume status are reliable and accurate when compared to measurements of a POCUS expert. The highest agreement is received when the image quality is good, especially for the automatic IVC and EF measurement tools.

POCUS operators, especially in the ICU, ED, and operating room, often operate under stress and time pressure and must repeat their patients’ assessment along the acute treatment. Automatic tools such as the ones described in this study have the potential to support eyeballing assessment of the LVEF, as well as more accurate measurements such as stroke volume and IVC collapsibility index, allowing easy and rapid measurements that are relatively more complicated when measured manually. These tools hold the potential to reduce measurement errors and variability, enhance performers’ self-confidence, and may lead to better clinical decisions. We believe that for less experienced POCUS operators for whom stroke volume and IVC measurements are still challenging, the automatic tools’ feedback can be reassuring and teaching at the bedside, where POCUS mentoring is often lacking.

The quality indicator can also be useful for technical improvement, regardless of the measurements that follow. Our study has shown good correlation agreement regarding image quality between a POCUS expert in both the EF auto tool and the IVC auto tool. This means that the auto-tools can differentiate high quality informative images from uninformative low-quality images with good accuracy for these views. Applying such tool in real time can potentially provide affirmation and defer from acquiring low quality images, saving time and effort. We believe that novice users especially may benefit from this quality score during examinations which will assist them to acquire better images and will assure them of the images taken.

Another conclusion that can be drawn from the results is that for ultrasound AI tools, image quality is fundamental for reliable automatic measurements. This is a challenge, as ultrasound, in general, and POCUS specifically, is a highly operator-dependent modality. These tools, if accurate, can be beneficial for novice users, but the basic need for an accurate automatic measurement is good image quality. This may be an obstacle for the less experienced POCUS operator. If the automatic tools are accurate only in the hands of the experts, then the utility of such tools is questionable. Therefore, other automatic real-time tools for image-acquisition support and image quality optimization, organs’ border automatic enhancement, or even totally automatic views acquisition are needed. Image acquisition and automatic measurement tools can be real progress in the POCUS learning process.

### 8.1. Study Strengths

We have performed an extensive literature search. As this study was conducted several studies which examine auto POCUS tools were published. One study assessed an artificial intelligence tool that measures the IVC collapsibility index for online open-access IVC videos and compared them to POCUS experts with good results [25]. This study was conducted digitally, analyzing videos, and did not include live patients. Another study compared the auto IVC tool in the GE Venue™ device to measurements by POCUS experts and POCUS novices with good results; however, only a five-patient study population was examined [26]. For EF, a study which studied the agreement between POCUS experts, POCUS novices and the GE auto EF tool was published with good results [27]. For VTI, one study studied the agreement between the LVOT-VTI automated tool and manual measurements made by the Venue™ device. The study included a small number of tests that were measured by the auto LVOT-VTI tool (n = 46) as a secondary objective of the research, and unlike our study, compared the auto tool’s measurements to measurements conducted by relatively novice operators. The study has shown good agreement between a manual examination and the auto LVOT-VTI tool in the Venue™ device [28]. All in all, the issue of real time POCUS measuring tools has not yet been thoroughly studied. Our design was planned to validate the automatic tools when compared to the “gold standard”—a highly experienced operator certified by the American National Board of Echocardiography. As far as we know, this is the first real-life clinical study to validate all three Venue™ automatic real-time cardiac measurement tools. Meaning this study is a validation study for the entire Venue™ Cardiac Auto tools module and can increase the confidence of operators when using the Venue™ device.

### 8.2. Study Weaknesses

This study’s sample size is relatively small. The performance of the ultrasound studies and the manual measurements were conducted by a single expert, although the readings were carried out blindly to the automatic results. Though the expert is certified by the American National Board of Echocardiography and is very experienced, we believe adding more POCUS examiners and image interpreters could have contributed to this study’s reliability. Further investigation must be conducted before coming to a clear conclusion regarding the automatic tools’ reliability. The tested subjects presented mainly non-pathological views, (most had normal LVSF by both reading methods) and few had a severe cardiac pathology (i.e., two with an LVEF of less than 40% in the auto EF analysis). We recommend conducting a more clinically diverse study involving more pathological views, including different readers and more POCUS performers. This will better assist in determining the generalizability of the accuracy of the automatic tools.

## 9. Conclusions

The accuracy of the automatic hemodynamic assessment tools, the rapid hands-free measurements, and real-time image quality feedback suggests that these automatic tools are beneficial for POCUS operators, novice, or experts. IVC, EF, and VTI real-time measurements can be measured rapidly when using these automatic tools, with a high degree of confidence that the results are accurate. Nevertheless, for accurate results, good image quality is needed. The EF tool should be tested on abnormal cardiac function. More studies with a larger population and a more diverse population of patients and operators are required. We believe this to be a first step in validating and, hopefully, integrating AI-based, real-time, automatic POCUS tools. More such tools should be developed for other ultrasound measurements.

## Figures and Tables

**Figure 1 jcm-12-01352-f001:**
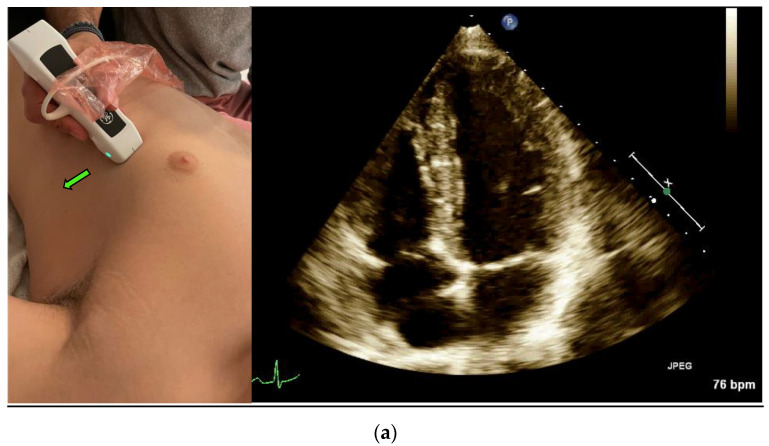
(**a**) Acquisition and anatomy of the Apical Four Chamber View. (**b**) Acquisition and anatomy of the Apical Five Chamber View (similar acquisition to Apical Four Chamber View, with a slight upward tilt of the probe). (**c**) Acquisition (probe point—green arrow) and anatomy of the Apical Two Chamber View.

**Figure 2 jcm-12-01352-f002:**
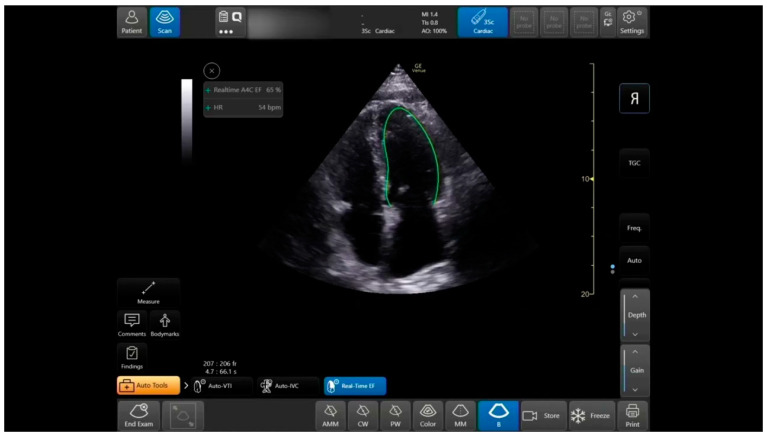
Auto EF tool: in Apical-4-Chamber-View: auto LV outlining in systole and diastole, measurements, and data display.

**Figure 3 jcm-12-01352-f003:**
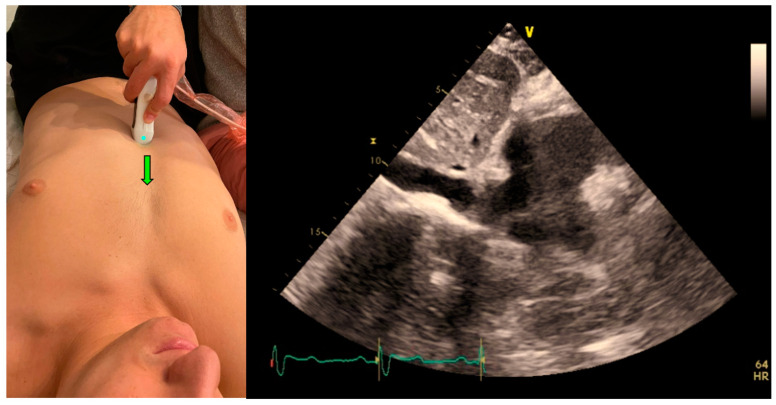
Acquisition (probe point—green arrow) and anatomy of IVC View.

**Figure 4 jcm-12-01352-f004:**
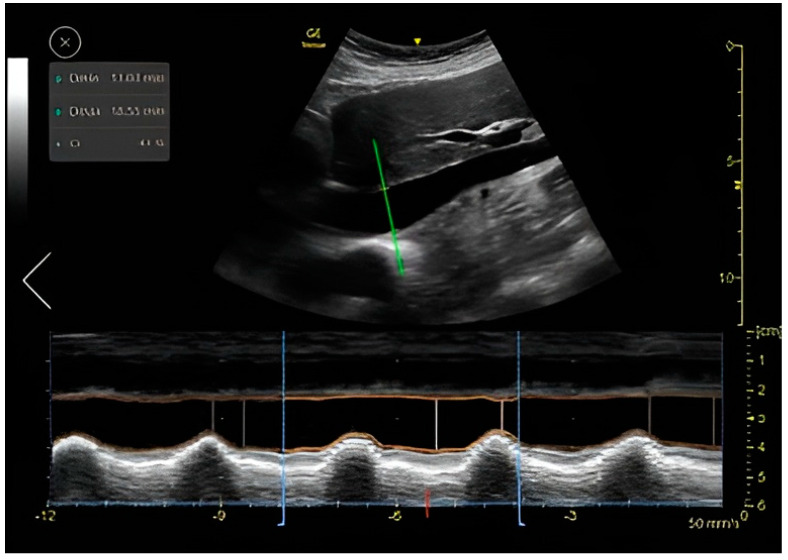
Auto IVC tool: auto marker placing, M-mode and data display.

**Figure 5 jcm-12-01352-f005:**
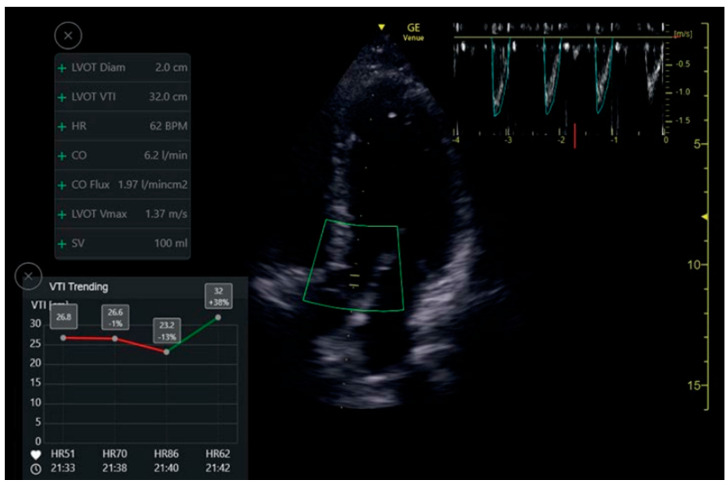
Auto VTI tool. When in Apical5Chamber-View. Auto marker placement, velocity pattern measurements and data display.

**Table 1 jcm-12-01352-t001:** Characteristics of patients.

Demographics and History	Real Time EF (N = 44)	Auto IVC Study (N = 37)	Auto VTI study (N = 32)
Sex % (n/N)			
Male	63% (28/60)	62% (23/37)	69% (22/32)
Age, y			
Mean (SD)	51 (17.9)	60.45 (15.61)	57.4 (18.5)
BMI, kg/m^2^			
Mean (SD)	26.8 (5.1)	26.98 (5.48)	27.1 (5.5)
Chronic Diagnoses % (n/N)			
Myocardial infarction	16% (7/44)	13.5% (5/37)	25% (8/32)
Congenital heart failure	13% (6/44)	10.81% (4/37)	16% (5/32)
Cerebrovascular disease	7% (3/44)	8.1% (3/37)	12.5% (4/32)
Chronic pulmonary disease	27% (12/44)	27.02% (10/37)	25% (8/32)
Diabetes	30% (13/44)	45.94% (17/37)	34.4% (11/32)
Renal disease	7% (3/44)	8.1% (3/37)	6.3% (2/32)

EF—Ejection Fraction, IVC—Inferior Vena Cava, VTI— Velocity Time Integral, SD— standard deviation.

**Table 2 jcm-12-01352-t002:** Ejection Fraction Stratified agreement.

Quality POCUS Expert	Quality by Auto EF	Agreement (Kappa)	*p* Value
overall agreement for all qualities		0.498	0.000
Green	Green	0.460	0.000
	Yellow	−0.034	0.923
Yellow	Green	−0.538	0.089
	Yellow	0.048	0.832

POCUS—Point Of Care UltraSound, EF—Ejection Fraction.

**Table 3 jcm-12-01352-t003:** IVC analysis—quality stratification and comparison between Auto LVOT-VTI tool and manual assessment by POCUS expert.

Quality- POCUS Expert	Quality- Auto IVC	Intraclass Correlationb	*p*
Overall Agreement		0.540	0.000
Green	Green	0.536	0.009
	Yellow	−1.314	0.848
Yellow		0.545	0.171

POCUS—Point Of Care UltraSound, IVC—Inferior Vena Cava, LVOT—left ventricular outflow tract, VTI—Velocity Time Integral.

**Table 4 jcm-12-01352-t004:** VTI analysis—quality stratification and comparison between auto LVOT-VTI tool and manual assessment by POCUS expert.

LVOT VTI VALUES						
Comparing Groups	Green		Yellow		Total	
	Physician (n = 26)	Automatic tool (n = 47)	Physician (n = 31)	Automatic tool (n = 10)	Automatic tool (n = 57)	Physician (n = 57)
Mean	19.1 ± 2.3	17.7 ± 4.3	20.1 ± 5	17.5 ± 5.4	19.5 ± 4.3	17.7 ± 4.5
Intraclass Correlation	0.655		0.914		0.825	
Sig.	0.024		*p* < 0.001		*p* < 0.001	

LVOT—Left Ventricular Outflow Tract, VTI—Velocity Time Integral.

## Data Availability

All supporting data are available upon request and pending the corresponding author’s (L.F.) approval.

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
