# Peer review of "Artificial Intelligence (AI) versus POCUS Expert: A Validation Study of Three Automatic AI-Based, Real-Time, Hemodynamic Echocardiographic Assessment Tools"

_jcm, 2023, doi:10.3390/jcm12041352_

Round 1

Reviewer 1 Report

Comments: The study “Artificial intelligence (AI) versus POCUS Expert: A validation study of three automatic AI-based, real-time, hemodynamic echocardiographic assessment tools.” is an interesting, important work. There is one of the first studies that tries to validate automatic real-time cardiac measurement tools for clinical practice. In the study three main parameters for emergency care are investigated (left ventricle ejection fraction, left ventricle outflow tract VTI, and collapsibility index of IVC).  The authors got an interesting and important data.

I believe it will be helpful for next clinical practice.  However, some questions need to be addressed:

Questions/Comments:

Major issue:

1.     Explain, please, how could it be: “132 clips from 44 pts. All clips were interpretable and included in the analysis”, and “53 observations were classified as preserved EF; 11 as moderately reduced EF, and 2 as severely reduced EF”, Results, Page 10. You analyzed only A4CH, as it was written in Methods, isn’t it?

53+11+2 ≠132;   53+11+2 ≠44. It isn’t clear.

2.     “The overall agreement between the automatic tool and the physician’s EF assessment was high (weighted Cohen's-Kappa 0.54; p< 0.001).” Page 11, 273

However, overall agreement in the Table2 is 0.498.

3.     Moreover, in the Discussion, Page 13, lines 291-295: “The Left Ventricle Ejection Fraction Automated Measuring Tool was reliable when good cardiac imaging quality is obtained. Most of the study population were patients with preserved LVEF, where the automatic tool was found to be especially reliable. We have found that the overall agreement between the Expert and machine is very high (0.825; p<0.001) and even higher for expert assessment of normal LV…” Which of the Kappa is real 0.54, 0.498 or 0.825 for left ventricle EF measuring? And where is “even higher” in the Results, that you mentioned in the Discussion.

Minor comment:

1.     Table 3. It isn’t clear, the value 0.366 and 0.536 both regarded to green/green, explain, please.

2.     Discussion. Page 13, Lines 282-285. “The Automatic tools inspected in this study are aimed to be used in clinical settings by clinicians. To Determine the true feasibility of implementing the automatic tools into everyday clinical use, they need to be assessed by clinicians on real patients in real clinical settings.” I think the main findings of the study could be formulated more precisely. Rephrase it, please.

3.     There is a repetition of the phases: Page 2, lines 56-58 and Page 3, 65-68.

4.     It would be interesting if you write the number of cases in Table 4.

Author Response

Major comment 1: Explain, please, how could it be: “132 clips from 44 pts. All clips were interpretable and included in the analysis”, and “53 observations were classified as preserved EF; 11 as moderately reduced EF, and 2 as severely reduced EF”, Results, Page 10. You analyzed only A4CH, as it was written in Methods, isn’t it? 53+11+2 ≠132;   53+11+2 ≠44. It isn’t clear.

Author's response:

Thank you for the important comment. The 132 clips were comprised of 2 groups of paired (similar) clips. We have added an explanation in the text.

Text after alterations:

"One hundred thirty-two LVEF clips from 44 patients were acquired from the A4C view. The clips were taken in pairs- a POCUS expert acquired a clip, and then without changing the probe position acquired a second, similar clip. Thus, two groups of paired clips were formed. All clips were interpretable and included in the analysis. Patients were primarily males, with a mean age of 55 ± 19.8 and a mean body mass index (BMI) of 26.8 ±5.3 (table 1). Of the first group of clips assessed by the Expert (n=66), 53 observations were classified as preserved EF (group 1), 11 as moderately reduced EF (group 2), and 2 as severely reduced EF (group 3)…" Page 10 lines 224-234

---

Major comment 2:

The overall agreement between the automatic tool and the physician’s EF assessment was high (weighted Cohen's-Kappa 0.54; p< 0.001).” Page 11, 273 However, overall agreement in the Table2 is 0.498.

Major comment 3:

Moreover, in the Discussion, Page 13, lines 291-295: “The Left Ventricle Ejection Fraction Automated Measuring Tool was reliable when good cardiac imaging quality is obtained. Most of the study population were patients with preserved LVEF, where the automatic tool was found to be especially reliable. We have found that the overall agreement between the Expert and machine is very high (0.825; p<0.001) and even higher for expert assessment of normal LV…” Which of the Kappa is real 0.54, 0.498 or 0.825 for left ventricle EF measuring? And where is “even higher” in the Results, that you mentioned in the Discussion.

Author's response:

Thank you very much for this extremely important comment. The results in the table are the correct result. However, due to a mistake there has been a shift in the results presented in writing. In the Auto-EF segment, for the 'good quality clips agreement' the result was written to be- 0.498 (the actual result for the overall agreement) when the true result is 0.460. For the 'overall agreement' the result was written to be-0.54 (the result for the above/below 50% EF agreement) when the actual result is 0.498. and in the discussion the overall agreement for the Auto-EF was written to be-0.825 (the actual result for the overall auto VTI interclass coefficient). We are very, very sorry for this honest mistake. We cannot emphasize enough that this was due to an error in the presentation of the results and not in the result themselves. The results presented in the tables have not changed.

Text after alterations:

  • "High agreement (weighted Cohen's-Kappa 0.460; p< 0.001) was found between the AI-based automatic EF tool and the Expert’s assessment when the test was conducted on high-quality clips…" Page 10 Line 232-234
  • "The overall agreement between the automatic tool and the physician’s EF assessment was high (weighted Cohen's-Kappa 0.498; p< 0.001, Table 2)." Page 11 lines 237-239
  • "We have found that the overall agreement between the Expert and machine is high (0.498; p<0.001) and even higher for expert assessment of normal LV…" Page 13 Lines 295-297

---

Minor comment 1:

Table 3. It isn’t clear, the value 0.366 and 0.536 both regarded to green/green, explain, please.

Author's response:

Thank you for the important comment. The table presents the results of an Interclass Correlation Coefficient test. The table before alteration included two results: the results for Average Measures (0.536) and one for Single Measures (0.366). After consulting with our statistician, we believe that the Single Measures test is irrelevant and created unnecessary confusion for the reader and was therefore removed.

Text after alterations:

Quality POCUS expert

Quality by Auto-EF

Agreement

(Kappa)

p value 

overall agreement for all qualities

0.498

0.000

Green

Green

0.460

0.000

Yellow

-0.034

0.923

Yellow

Green

-0.538

0.089

Yellow

0.048

0.832

---

Minor comment 2:

Discussion. Page 13, Lines 282-285. “The Automatic tools inspected in this study are aimed to be used in clinical settings by clinicians. To Determine the true feasibility of implementing the automatic tools into everyday clinical use, they need to be assessed by clinicians on real patients in real clinical settings.” I think the main findings of the study could be formulated more precisely. Rephrase it, please

Author's response:

Thank you for the important comment. This sentence was not intended to present the main finding, but rather to emphasize it's importance. We completely agree it contained an excessive amount of the word 'clinic'. The sentence was rephrased accordingly.

Text after alterations:

"The Automatic tools inspected in this study are aimed to be used bedside in a clinical setting. To Determine the true feasibility of implementing the automatic tools into everyday clinical use, they need to be assessed on real patients in a real clinical setting." Page 13 Lines 285-287

---

Minor comment 3:

There is a repetition of the phases: Page 2, lines 56-58 and Page 3, 65-68.

Author's response:

Thank you for the important comment. The repetition was fixed.

Text after alterations:

"Moreover, manual accurate measurements are often complicated and time-consuming(11). For example, the Simpson method for measuring EF requires the operator to manually trace the endocardial border in systole and diastole in at least two views. Accurate and time-efficient real-time automatic tools could improve urgent decision-making and, therefore, patient care". Page 2 Lines 55-59

---

Minor comment 4:

It would be interesting if you write the number of cases in Table 4.

Author's response:

Thank you for the important comment. Done.

Text after alterations:

See table 4.

Reviewer 2 Report

The authors present a very interesting article about the accuracy and effectiveness of the automatic tools integrated into one of the echo devices.  What is particularly important is that those tools are used in pocus examination, which by definition should be quickly and easily performed. It is performed not only by a cardiologist but also by anesthesiologists, and cardio surgeons.

-       The major limitation is the comparison of the results with only one of the experts. It would be more reliable

-       The 3-degree scale of image quality is not precise. Do we have any correlation with obesity or general condition?

-       One person was obtaining the images, and there was no second examination by the second observer

-       Was the EF evaluated by the expert at the workstation or GE Venue? Expert evaluation of the grade 1 image quality may also have an important bias with the gold standard. Do you have data from the semi-automatic evaluation of the workstations?

-       Only two patients with the EF <40% is an important limitation of the study

-       In the limitations section it should be stated that for the cardiac output, we need invasive measurements as a gold standard

-       What was intraobserver variability for the experienced physician?

Author Response

comment 1: The major limitation is the comparison of the results with only one of the experts. It would be more reliable

comment 3: One person was obtaining the images, and there was no second examination by the second observer

Author's response:

Thank you for these two important comments. This is very true. A reference to these issues was added to the limitations section.

Text after alterations:

" The performance of the ultrasound studies and the manual measurements were conducted by a single Expert, although the readings were done blindly to the automatic results. Though the expert is, as mentioned, certified by the American National Board of Echocardiography and is very experienced, we believe adding more POCUS examiners and image interpreters could have contributed to this study's reliability." Page16 Lines380-385

comment 2: The 3-degree scale of image quality is not precise. Do we have any correlation with obesity or general condition?

Author's response: Thank you very much for this important comment. Unfortunately, we only examined the image quality scale as a secondary objective and did not study the correlation it has with different patient characteristics. 

Comment 4: Was the EF evaluated by the expert at the workstation or GE Venue? Expert evaluation of the grade 1 image quality may also have an important bias with the gold standard. Do you have data from the semi-automatic evaluation of the workstations?

Author's response: Thank you for the important comment. All evaluation was done bedside with the GE Venue device. Therefore, we do not have any data from the workstation.

Comment 5: Only two patients with the EF <40% is an important limitation of the study

Author's response: Thank you for the important comment. We have added a sentence that addresses this issue in the study limitations section.

Text after alterations: "The tested subjects presented mainly non-pathological views, (most had normal LVSF by both reading methods) and few had a severe cardiac pathology (i.e., two with an LVEF of less than 40% in the auto EF analysis)." Page 16 Lines 385-388

Comment 6: In the limitations section it should be stated that for the cardiac output, we need invasive measurements as a gold standard

Author's response: Thank you for the comment. We kindly disagree with this comment. We do not presume to compare the Auto-Tools to the actual 'Gold Standard' of each condition (for example, the 'Gold standard' for ejection fraction is the CMR), but rather to the POCUS 'Gold Standard', which is the most accurate POCUS examination, in this case- a highly trained and experienced POCUS expert. Therefore, we believe that referring to each condition's 'Gold Standard' can potentially confuse the reader.

-

Comment 7: What was intra-observer variability for the experienced physician?

Author's response: Thank you for the important comment. As mentioned, each clip was acquired and examined once, and therefor we could not measure intra observer variability as there was no second round of examinations.  

Round 2

Reviewer 1 Report

Authors have addressed all the issues raised before.